# Radioresistance of Non-Small Cell Lung Cancers and Therapeutic Perspectives

**DOI:** 10.3390/cancers14122829

**Published:** 2022-06-08

**Authors:** Mathieu Césaire, Juliette Montanari, Hubert Curcio, Delphine Lerouge, Radj Gervais, Pierre Demontrond, Jacques Balosso, François Chevalier

**Affiliations:** 1Department of Radiation Oncology, Centre François Baclesse, 14000 Caen, France; m.cesaire@baclesse.unicancer.fr (M.C.); j.balosso@baclesse.unicancer.fr (J.B.); 2UMR6252 CIMAP, Team «Applications in Radiobiology with Accelerated Ions», CEA—CNRS—ENSICAEN—Université de Caen Normandie, Campus Jules Horowitz, Bd Henri Becquerel, BP 55027, CEDEX 05, F-14076 Caen, France; juliette.montanari@ganil.fr; 3Department of Medical Oncology, Centre François Baclesse, 14000 Caen, France; h.curcio@baclesse.unicancer.fr (H.C.); d.lerouge@baclesse.unicancer.fr (D.L.); r.gervais@baclesse.unicancer.fr (R.G.); p.demontrond@baclesse.unicancer.fr (P.D.)

**Keywords:** non-small-cell lung cancer, cancer stem cell, cancer mutation, radiotherapy, immunotherapy, proton irradiation

## Abstract

**Simple Summary:**

The poor survival of unresectable locally advanced stage non-small cell lung cancer is due to the resistance to chemoradiotherapy and local/distant relapses. However, with the advent of new drugs, it has become possible to improve the prognosis of patients with stage III NSCLC harboring certain genetic mutations. Herein, we review new therapeutic strategies to overcome this radioresistance with drugs targeting cancer stem cells/specific mutations or new radiotherapy modalities.

**Abstract:**

Survival in unresectable locally advanced stage non-small cell lung cancer (NSCLC) patients remains poor despite chemoradiotherapy. Recently, adjuvant immunotherapy improved survival for these patients but we are still far from curing most of the patients with only a 57% survival remaining at 3 years. This poor survival is due to the resistance to chemoradiotherapy, local relapses, and distant relapses. Several biological mechanisms have been found to be involved in the chemoradioresistance such as cancer stem cells, cancer mutation status, or the immune system. New drugs to overcome this radioresistance in NSCLCs have been investigated such as radiosensitizer treatments or immunotherapies. Different modalities of radiotherapy have also been investigated to improve efficacity such as dose escalation or proton irradiations. In this review, we focused on biological mechanisms such as the cancer stem cells, the cancer mutations, the antitumor immune response in the first part, then we explored some strategies to overcome this radioresistance in stage III NSCLCs with new drugs or radiotherapy modalities.

## 1. Introduction

Lung cancer was the leading cause of cancer mortality worldwide in 2015 with 1.6 million deaths [1]. Non-small cell lung cancer (NSCLC) is the most common histological form accounting for 80% of all lung cancers. Among patients with NSCLC, approximately 35% are at a locally advanced unresectable stage with the overall survival at 5 years remaining low at approximately 10 to 15% despite treatment with radiotherapy combined with chemotherapy [2]. Recently, adjuvant Durvalumab immunotherapy improved overall survival for unresectable stage III NSCLC patients responding to chemoradiotherapy [3]. However, even in this selected population, the overall survival at 3 years was 57% [4].

The TNM staging is the current main prognosis factor of survival in stage III unresectable NSCLC treated by chemoradiotherapy but others factors such as WHO performance status, gender, number of positive lymph node stations, gross tumor volume, clinical T stage, chemotherapy, radiation therapy overall treatment time, and radiotherapy dose could be involved in the prognosis [5].

The mutational status in driver genes such as epidermal growth factor receptor (EGFR) mutations, Kirsten rat sarcoma viral oncogene (KRAS) mutations, or anaplastic lymphoma kinase (ALK) translocations or mutations could drive the prognosis for stage III unresectable NSCLC treated by chemoradiotherapy [6,7,8,9,10]. However, despite these potential mutational status prognoses, at our knowledge, there are few biological mechanisms clearly recognized explaining radiotherapy failure for stage III NSCLC.

In the recent decades, cancer stem cells or cancer initiator cells (CSCs) have been described as a possible origin of aggressive characteristics of cancers with properties of self-renewal, metastasis formation, and resistance to treatments [11].

In this review, we have focused in the first part on the biological resistance mechanisms such as cancer stem cells, mutational status, and the immune response, then we have reported some strategies that have been already explored or been under investigation to overcome radioresistance in stage III unresectable NSCLCs.

## 2. Molecular Radioresistance Mechanisms

### 2.1. Radioresistance Linked to Hypoxia, Cancer Stem Cells, and Epithelial–Mesenchymal Transition

CSCs are a subpopulation of cancer cells that have self-renewal and tumor initiating properties, and play an important role in metastasis, tumor relapse, and resistance to treatments such as chemotherapy, radiotherapy, and immunotherapy. In a clinical study, patients with locally advanced stage II or III lung cancer with cancer stem cell markers, such as the Nestin marker, had poorer overall survival [12]. The prognostic role of genes significantly overexpressed in tumorspheres was evaluated in an NSCLC cohort (CDKN1A, SNAI1, and ITGA6 were found to be associated with prognosis and used to calculate a gene expression score, named the CSC score): survival analysis showed that patients with a high CSC score had a shorter overall survival (OS) [13].

The resistance to radiation therapy in NSCLC could be caused by a selected radioresistant population of CSCs as suggested by studies on cell lines or animal models. Indeed, X-ray photon irradiation resulted in an increase in cancer stem cells markers (CD44, CD133, OCT4, SOX2, and NANOG) [14,15] and increased the pool of cancer stem cells in xenograft mouse models [16]. Cancer stem cells develop particularly in hypoxic tumor niches. For example, in vitro models of the A549 cell line cultured in hypoxic conditions have been able to show an increase in markers attributable to cancer stem cells [17]. Hypoxia is defined as a lower oxygen pressure (usually between 1 and 10 mmHg) than in healthy tissue (usually between 40 and 60 mmHg) and plays a major role in the tumor microenvironment that can confer resistance to chemotherapy and radiotherapy [18]. Hypoxia regions in lung cancers could be found using Positron Emission Tomography (PET) imaging with an 18F-fluoromisonidazole (FMISO) hypoxic tracer and may present radioresistance [19]. These hypoxia regions might therefore be targeted by some types of local dose escalation in radiotherapy or molecular therapies in association with radiotherapy.

Cancer progression is often associated with the epithelial–mesenchymal transition (EMT) process. Several studies have demonstrated the implication of the EMT in the proliferation and migration of epithelial cancer cells, such as in NSCLCs, resulting in metastatic properties and resistance to treatments [20]. EMT is a biological process where epithelial cells pass to a mesenchymal state. During EMT, epithelial cells lose their apical–basal polarity and their cell–cell junctions, as well as their cytoskeletal organization and adhesion being altered, inducing cell plasticity and allowing them to leave the epithelial tissue to migrate to other tissues. EMT is an essential biological program for the development during embryogenesis, the mesoderm development during gastrulation, tissue morphogenesis, and wound healing [21]. In addition, studies have been reported that cells undergoing EMT exhibit molecular alterations by the decrease in epithelial markers such as E-cadherin, ZO-1, and occludin, and the increase of markers associated with the mesenchymal state such as N-cadherin, vimentin, fibronectin, and fibroblast-specific protein 1. EMT is driven by transcriptions factors (SNAIL, ZEB, SLUG, or TWIST), miRNAS, and epigenetic regulators that are involved in cell plasticity during tumorigenesis [22,23]. EMT is induced by the activation of several intracellular signaling pathways such as TGFβ, WNTs, NOTCH, and EGFR [22]. The tumor microenvironment can also be involved in the EMT program by paracrine signals that promotes tumor progression and metastasis [22]. Therefore, targeting EMT and signaling pathways promoting EMT could be a great strategy in NSCLC treatment such as TIF1γ, a TGFβ signaling regulator, that could act as a tumor metastasis suppressor in NSCLC by inhibiting EMT induced by TGFβ [24]. Moreover, cancer cells that have undergone EMT show stem cells-like characteristics and a resistance to therapeutic treatments [20]. CSC markers in association with EMT markers have been reported in NSCLCs with the worst prognosis [25,26]. Radiation by photons X induced EMT markers (SNAIL and PDGFR-beta) in vitro [15] and several studies showed the association of EMT and radioresistance in NSCLC [15,27,28,29]. For example, EMT was induced by miR-410 with a radioresistance via the PTEN/PI3K/mTOR axis in NSCLC, so miR-410 could be used as a biomarker or therapeutic agent in NSLCL [29]. Other signaling pathways could be targeted to reverse resistance in NSCLC such as Wnt/*β*-catenin [30]. However, despite much research on EMT in order to find new approaches to control CSC-associated drug resistance and the EMT program, EMT mechanisms are not fully understood and more studies need to be carried on. 

### 2.2. Radioresistance Linked to Mutational Status and Therapeutic Approaches

Radiotherapy is one of the most common treatments for NSCLC, and the tumor sensitivity to radiotherapy may affect individual prognoses of NSCLC. However, predictable signatures related to the radiotherapy response are still limited [31]. The importance of upregulation, enhanced activation, and critical signature of the PI3K/Akt/mTOR pathway are well documented. Somatic mutations involving various parts of the display cascade and gene enhancement have been shown in various cancers. Changes in PIK3CA have been observed in 36% of hepatocellular cancers, 26% of breast, and 26% of colon cancers. Low levels of PIK3CA mutations are seen in ovarian, gliomas, stomach, and lung cancers. In NSCLC, the PI3K/Akt/mTOR pathway has been significantly implicated in both tumorigenesis and disease progression. Several inhibitors of PI3K, Akt, and mTOR are currently being developed and are in various stages of pre-clinical research and phase I clinical trials in NSCLC [32].

The mechanism of action of EGFR is central to cell proliferation and survival [33]. Ligand activation of EGFR by epidermal growth factor or other ligands, leads to the activation of several survival signaling pathways, including mitogen-activated protein kinase, phosphoinositide 3-kinase (PI3K)/AKT, and signal transducers and activators of transcriptional signing cascades. NSCLC have been reported to show mutations in many oncogenes and tumor suppressor genes, including EGFR, KRAS, and tumor protein 53 (TP53). NSCLC cells express EGFR and its derivatives, which play a key role in the pathogenesis of lung cancer [33], and therefore, the cell blockade of the EGFR signaling represents a promising cancer treatment strategy. Several studies have shown strong preclinical and clinical evidence supporting the potential of targeting EGFR signaling to improve the antitumor activity of ionizing radiation [34]. EGFR mutations have appeared to be more sensitive to radiation therapy [6] than other mutations, but the history of EGFR-mutated tumors is linked to frequent metastatic evolution. Indeed, in unresectable stage III EFGR-mutated NSCLC patients, failure to radiation therapy treatment was related to spread out disease [7]. EGFR is a trans-membrane glycoprotein with an extracellular part that binds growth factor proteins, and an intracellular part that have a tyrosine kinase domain. EGFR activation induces cellular proliferation by a signaling pathway [35]. EGFR mutations leading to the overexpression of EGFR proteins have been reported to induce carcinogenesis in NSCLC [36]. EGFR tyrosine kinase inhibitors (TKI) such as Osimertinib have been mainly used in metastatic patients, improving patient survival [37] with good responses in brain metastasis [38]. Recently, the adjuvant treatment of EGFR TKI Osimertinib for resected locally NSCLCs have been reported to improve overall survival and disease-free survival [39]. Osimertinib appeared particularly efficient to prevent metastatic evolution in the central nervous system as compared with other TKIs or chemotherapy [40]. Adjuvant therapy with EGFR TKI, and particularly Osimertinib, is currently under investigation in clinical trials to improve survival and prevent metastatic evolution in EGFR-mutated locally unresectable stage III NSCLC patients treated by chemoradiotherapy [41]. The first generation of EGFR TKIs such as Gefitinib or Erlotinib, in concomitant with chemoradiotherapy, failed to improve efficacity in unresectable stage III NSCLCs [42,43]. The third generation of EGFR TKIs, such as Almonertinib, might be more efficient in concomitant with chemoradiotherapy and are currently being investigated in phase 2 clinical trials (NCT04636593, NCT04952168).

KRAS mutations are involved in carcinogenesis with a frequency of 25–30% in NSCLC [44,45]. KRAS protein is an intracellular submembrane GTPase enzyme that transmits the signal from the EGFR located on the surface of cells by the hydrolysis of GTP into GDP. This signal induces a transduction cascade leading to cell division. A simple mutation of the KRAS gene (most often at codons 12 or 13) can induce constitutive activation of the KRAS protein independently of the EGFR, inducing tumor growth [44]. KRAS mutations are involved in resistance to radiation therapy [8]. In a retrospective study, patients with stage III NSCLC treated with chemoradiotherapy had a poorer response to treatment in case of KRAS mutation [9]. In NSCLC patients with brain metastases treated by encephalic irradiation, KRAS mutations confer radioresistance [6]. The mechanisms of radioresistance of mutated KRAS lung cancer cells are still relatively unknown. Radioresistance in mutated KRAS lung cancers may be related to CSC properties [46]. In vitro studies and pathology analysis of resected cancers showed that the KRAS mutation induced an overexpression of RAD51, which is a recombinase protein involved in the repair of DNA damage (by the homologous recombination mechanism), and so could lead to resistance to platinum salt chemotherapy and radiotherapy [47]. The KRAS mutation via an EGFR-dependent pathway promotes a chromatin condensation inducing radioresistance in NSCLC in vitro and in vivo [46,48]. TKI targeting KRAS G12C mutated NSCLCs have been reported to induce a good response as a second line treatment for metastatic patients [49]. Targeting KRAS proteins in mutated NSCLCs might be interesting to improve the response to radiotherapy. Some co-mutations occur frequently with KRAS mutations. One of these mutations, the STK11 mutation has been reported to induce resistance of NSCLCs to treatment and stem-cell-like properties [50] and might be also a target of interest.

### 2.3. Radioresistance Linked to the Modulation of the Immune Response

The Programmed cell Death protein 1 receptor (PD-1) is a T cell receptor that inhibits the cytotoxic activity of these cells, avoiding autoimmune diseases. PD-1 interacts with two ligands, PDL-1 and PDL-2, present on several types of cells membranes, such as various immune cells, mesenchymal support cells, and vascular cells. The upregulation of PDL-1 by tumor cells confers resistance to the immune system [51]. To reverse this immune escape, anti PD-1/PDL-1 have been clinically developed in many types of tumor, such as NSCLC [52]. PD-L1 expression was increased after conventionally fractionated radiation in several studies [53,54,55], with an impact on the antitumor response to radiation [55]. Radiotherapy may up-regulate PD-L1 expression through the phosphoinositide 3-kinase/AKT and signal transducer and activator of transcription 3 pathways. PD-L1 has been reported also to stimulate cell migration and facilitate the epithelial–mesenchymal transition process [55]. This radioresistance pathway could be overcome with anti PD-1 immunotherapies. Indeed, adjuvant anti-PD-L1 immunotherapy such as Durvalumab after chemoradiotherapy in locally advanced NSCLCs improved survival [4].

## 3. Therapeutic Approaches to Overcome Radioresistance in NSCLC

### 3.1. Dose Escalation and Dose Painting in Radiotherapy

Conventional radiotherapy associated with concomitant chemotherapy based on platinum is the standard treatment for inoperable stage III NSCLC patients. Radiotherapy in this indication usually delivers 60 to 66 Gy in 30 to 33 daily fractions of 2 Gy [2]. A phase III trial in unresectable stage III NSCLCs tried to overcome radiation resistance by escalating the dose at 74 Gy with an unexpected result due to toxicity: survival was worse in the patient group treated with 74 Gy than in the patients group treated with 60 Gy [56]. An escalating dose (from 70 to 90 Gy) increased cardiac toxicity and reduced survival despite the antitumor benefit [57]. However, clinical studies have reported contradictory results and some studies showed better survival with an individually adapted escalating dose (84–102.9 Gy) approach based on predictive irradiated lung tissue volumes [58]. Despite this personalized approach in escalating dose, survival remains poor with high toxicity. Heterogeneous dose distribution guided by predictive markers of local radioresistance in the tumor might represent the future of escalating schemes of radiotherapy. Several clinical studies have investigated the efficacy of a limited escalating dose (boost) guided by PET imaging (RTEP studies). A RTEP7 study is currently investigating the use of a boost up to 74 Gy, limited to the tumor region remaining [^18^F]FDG hypermetabolic after 42 Gy (NCT02473133). This approach is named dose painting or biologically-guided dose painting. [^18^F]FMISO appeared to have a different distribution without any correlation with [^18^F]FDG and might conduce to different approaches of targeting hypoxia tumor regions [59,60].

### 3.2. Concomitant and New Radiosensitizing Treatments

Since now, only chemotherapies based with platin salts are the concomitant treatments associated with radiation therapy to treat unresectable stage III NSCLCs [2,61]. The additive or synergistic effect as a radiosensitizer of platin salts is debated and the radiosentive effect could be explained with implication of the ATM pathway [62]. Cisplatin was reported to radiosentize in vitro in an NSCLC cell line such as H460 and to have no effect on other NSCLCs such as A549 [62]. Several mechanisms could explain platin salt resistance in NSCLCs such as the reduced intracellular accumulation of cisplatin, the enhanced drug inactivation by metallothionine and glutathione, the increased repair activity of DNA damage, and the altered expression of oncogenes and regulatory proteins [63]. New therapies targeting DNA damages repair pathways such as poly(ADP-ribose) polymerase inhibitors (PARPi) could enhance the response to chemoradiotherapy in NSCLC. The poly(ADP-ribose) polymerase (PARP) are proteins implicated in the recognition and repair of DNA damages [64]. So PARP inhibitors induce cell mortality by the accumulation of DNA damages and act as a radiosensitizer in many in vitro and in vivo studies on several cancer cell lines including NSCLC [65]. The PARPi radiosentisation ratio was comprised between 1.1 and 1.62 in normoxia (21% O_2_) and could reach 2.87 in hypoxia conditions (1% O_2_) in xenograft models [66]. PARPi and platin salts could also synergize in NSCLC [67]. So PARPi might represent an interesting association in chemoradiotherapy for unresectable stage III NSCLCs, as suggested by a phase 2 clinical trial with Veliparib [68]. Furthermore, PARPi alone or in association with radiation therapy could activate the antitumor immune response and synergize with immunotherapies [69]. The association between a PARPi such as Niraparib and radiation showed the activation of the antitumor immune response in NSCLCs with an increase in CD8+ T lymphocytes, the activation of the STING/TBK1/TRF3 pathway, and the expression of chemokines such as CCL5, CXCL10, and cytokines such as interferon β [70]. However, PARPi can induce hematologic toxicity such as neutropenia, limiting its association with chemotherapy [71]. Nanoparticles administration of PARPi can improve diffusion to the tumor due to their passive targeting ability on tumor tissue that has an enhanced permeation and retention effect [72,73,74]. Therefore, PARPi nanoparticles could enhance the radiosensitizing effect in NSCLC [75].

### 3.3. Concomitant Immunotherapy

Radiotherapy can induce either immunosuppression or an antitumor immune response depending on various parameters. Radiotherapy in a large field including lymph nodes and vessels has been reported to induce lymphopenia in NSCLCs, depending of the radiation dose and the radiation volumes [76,77,78]. This radiation impact on the T lymphocytes, particularly with the daily fraction scheme of 2 Gy, can induce more failure in disease control due to an immunosuppressive effect [79]. Hypofractionated radiotherapy with high dose per fraction (6 to 10 Gy) such as stereotactic radiotherapy for early stage NSCLCs, appeared to stimulate the immune response with immunogenic cell death better than conventional radiotherapy with 2 Gy daily fractions [80]. Hypofractionated radiotherapy in unresectable stage III NSCLCs delivers 55 to 66 Gy in 20 to 24 fractions of 2.75 Gy for example [81,82]. This schedule of radiotherapy could not induce the same effect as stereotactic radiation that can spare more healthy tissues. Furthermore, conventional radiotherapy could upregulate PD-L1 in NSLCs [53,54,55]. Concomitant immunotherapy might counterbalance this immunosuppressive effect of chemoradiotherapy, in particular anti PD-1 immunotherapy, as suggested in NSLC mice models [53,54,55]. AIRING (Accelerated Radio-Immunotherapy for Lung Cancer) is a phase II clinical trial that is currently investigating the potential association Nivolumab (an anti PD-1 immunotherapy) in association with radiotherapy for patients who are not eligible for concomitant chemotherapy (NCT04577638). However, concomitant immunotherapy, in particular anti CTLA-4 immunotherapy (Ipilimumab), can increase pulmonary toxicity [83]. Consolidation anti PD-1 immunotherapy (Pembrolizumab) did not increase toxicity in a phase 2 trial [84] and consolidation anti PD-L1 immunotherapy (Durvalumab) was well tolerated despite a slight increase in pulmonary toxicity in the PACIFIC phase 3 trial [3], encouraging the use of immunotherapy targeting the PD-1/PD-L1 axis with chemoradiotherapy. Many trials currently investigate various protocols of checkpoint inhibitor immunotherapies (anti PD-1, anti PD-L1, and anti CTLA-4) in induction, concomitant, or in consolidation with chemoradiotherapy in unresectable stage III NSCLCs, as shown in Table 1.

### 3.4. Adjuvant Immunotherapy

Immunotherapy can also be used as a maintenance therapy after the end of the chemoradiotherapy treatment. One of the first trials assessing maintenance therapy with an immunotherapeutic agent is the cohort number four of the NCT00455572 trial, in which patients presenting with cancer/testis antigen MAGE-A3-positive NSCLCs received intramuscular injections of MAGE-A3 with the immunostimulant agent AS15. MAGE-A3 is considered cancer-specific because the physiological cells expressing it, i.e., spermatogonia and trophoblasts, cannot present epitopes because of the lack of major histocompatibility complexes on these cells’ membranes. After injections, all the patients of the cohort were seropositive for MAGE-A3-specific antibodies vs. 1/12 patients at baseline and 5/6 and 2/6 assessable patients that had MAGE-A3-specific CD4+ and CD8+, respectively, T-cells. Efficacy has not been reported for this trial but a lack of efficacy using the same agent in a different setting mandated the discontinuation of the investigations of this therapeutic [85].

Tecemotide, a synthetic lipopeptide derived from the mucin 1 (MUC1) sequence, was assessed as a maintenance therapy for MUC1-positive NSCLCs after chemoradiotherapy, as it was shown to induce a T-cell response in preclinical models and in patients. The START trial (Stimulating Targeted Antigenic Response To non-small-cell lung cancer) was a phase 3 trial that randomized MUC1-positive NSCLC patients with at least a stable disease after the initial chemoradiotherapy between maintenance tecemotide vs. a placebo. The trial failed to achieve its primary endpoint with a median overall survival (OS) of 25.6 months for tecemotide versus 22.3 months for placebo (HR = 0.88; *p* = 0.123) [86].

Durvalumab, an anti-PD-L1 antibody, has been assessed in the PACIFIC study, a phase 3 trial evaluating maintenance with durvalumab for 1 year versus a placebo. The trial showed improved progression free survival (PFS) (16.8 vs. 5.6 months) [3] and OS (28.3 vs. 16.2 months) [87] favouring the durvalumab arm. These results were confirmed with an updated 4 years OS rate of 49.6% for the durvalumab arm versus 36.3% for the placebo arm [88]. Subgroup analysis showed that PDL-1-positive patients tend to derive more benefit from this treatment than PDL-1-negative patients. Durvalumab maintenance therapy is now part of the standard of care for stage III NSCLC patients who have at least a stability after chemoradiotherapy.

Other anti-PD-1 agents such as Nivolumab and Pembrolizumab have been tested in smaller phase 2 trials. Nivolumab has been assessed in the NICOLAS trial in concomitance with chemoradiotherapy and in maintenance for a maximum of 1 year if the patient did not progress at the end of it. Seventy-nine patients were enrolled, the median PFS and OS were 12.7 months and 38.8 months, respectively [89]. A randomized phase 3 study versus a placebo was designed but only eight patients were randomized because of the publication of the PACIFIC study and giving a placebo to the patients was deemed unethical [90]. Pembrolizumab has been assessed in the HCRN LUN14-179 trial as a maintenance therapy for a maximum of 1 year after chemoradiotherapy for patients with stage III NSCLC. A total of 93 patients were enrolled, the median PFS and OS were 18.7 months and 35.8 months, respectively (NCT02343952). There is a randomized versus placebo phase 2 trial ongoing in Italy (NCT03379441).

Immunotherapies can also be used in association to try to prevent secondary resistance to immunotherapy maintenance. COAST is a phase 2 trial that randomized in a 1:1:1 manner patients between durvalumab alone or in association with oleclumab, an anti-CD73 monoclonal antibody, or monalizumab, an anti-NKG2A monoclonal antibody, for up to 12 months of treatment. Median PFS was 6.3 months for durvalumab alone, 15.1 months (HR = 0.65) for the durvalumab + monalizumab arm, and not reached (HR = 0.44) for durvalumab + oleclumab [91]. These first results have to be analyzed with caution as the durvalumab alone arm compares poorly with the results of the PACIFIC trial.

Finally, PD-1 inhibitors can be used in association with anti-TIGIT antibodies. Ongoing phase 3 trials such as KEYVIBE-006 (NCT05298423) and SKYSCRAPER-03 (NCT04513925) compare durvalumab with the combination of pembrolizumab and vibostolimab or the combination of atezolizumab and tiragolumab, respectively, for up to 12 months of treatment.

### 3.5. New Irradiation Techniques: Hadrontherapy

One of the simplest ways to combat tumor radioresistance is of course to increase the dose delivered to tumors. But toxicity, as mentioned above, represents a limit that is difficult to overcome. Nevertheless, this approach has already proven its usefulness, as shown by the use of techniques that have ballistic qualities that allow higher doses to the target thanks to more precise conformation capacities and an improved protection of nearby organs at risk. A first step has been taken in this field by the generalization of IGRT with intensity modulation (IMRT and VMAT), which has made it possible to climb a step in terms of the delivered dose by increasing it by approximately 10 to 20% (increased from 55–60 Gy to 66 Gy for example) for the same or even lower toxicities [56].

But other advances based on the physical qualities of radiation are possible. Thus, we can use radiation that has no exit beam such as charged particles beams, as protontherapy, which represents a new step in the escalation of tumor doses particularly studied in the context of lung tumors [92]. Some clinical studies, using prontherapy in unresectable stage III NSCLCs, completed or currently recruiting, have been reported in Table 2.

An additional step consists of matching the ballistic capabilities of the radiotherapy devices and the tumor biology [93]. Thus, the possibility of specifically defining the radioresistance characteristics of tumors, in particular by the presence of hypoxia zones, makes it possible to deliver localized dose increments in hypoxic/radioresistant zones by the dose painting technique mentioned above. But this technique is supposed to know at each session (therefore each day) where these areas of hypoxia are located in the tumor and to adapt the treatment to them in real time. The means to do this do not yet exist but an approach using a combination of a PET camera and a Linac could be interesting [94].

Finally, approaches based on even more innovative irradiation methods to force tumor radioresistance in a less discriminating way than by the daily imaging of hypoxia or other markers of radioresistance, could come from the use of hadrontherapy. Indeed, hadrontherapy introduces an additional notion of relative biological efficacy that is higher than X-rays in their ability to destroy tumors [95].

Actually, all radiotherapy techniques using ionizing radiation are based on the same mechanisms. Namely, the production of powerful oxidizing radicals by the radiolysis of water in tissues and cells. These radicals are responsible for most of the molecular damages to DNA that are essentially, regarding the types to be considered for cell lethality, complex damages often summarized by the term of DNA double-strand breaks.

Nevertheless, a certain number of distinctions at the nanometric scale can be made between the different types of radiations, particularly in terms of ionization density along the trajectories crossing the tumors. X-rays (photons) produce very diffuse ionizations causing few complex damages relatively distant from each other, producing very diffuse oxidative stress that effectively stimulates the cellular defenses. Thus, as mentioned previously, X-rays can stimulate EMT and reinforce the CSC phenotype and therefore induce their own radioresistance [96].

Conversely, ions such as protons (protontherapy) or heavier ions (hadrontherapy) obtained from carbon, oxygen, or helium atoms will cause very high ionization densities (100 to 1000 times more important than X-rays [97]) bringing a lot of complex damages close to each other in the very interior of the cell nucleus leading to irreversible damages capable of exceeding the radioresistance capacities of the cells. In addition, the grouping of ionizations along the trajectories, less numerous at an equal dose for the ions, considerably reduces the intracellular oxidative stress [96]. Thus, for cells surviving irradiation by ions, the cellular defense reactions elicited by irradiation will be much weaker than for X-rays. These two characteristics: greater physical efficiency and less capacity to stimulate tumor radioresistance, mean that ion therapy may have a greater efficacy on radioresistant tumors compared to X-rays. It is very likely that the very characteristics of tumor stem cells can explain this property and one can imagine that the in-depth analysis of tumors, possibly their greater or lesser richness in CSCs, can also ultimately be a guiding factor towards irradiation by ions rather than by photons.

Thus, faced with the various oncological situations of radioresistance, hadrontherapy represents a hope of progress in terms of the locoregional treatment of conceptually rather simple implementation.

However, the progression of the local control can always be discussed if the tumor evolution remains strongly marked by the metastatic evolution. Nevertheless, it should be kept in mind that local control remains essential for any hope of a cure, and that the primary tumor represents a cellular reservoir capable of producing variants resistant to successive systemic treatments. In addition, advances in systemic treatments and even the synergies between these systemic treatments and radiotherapy, in particular as for the forcing of the immune checkpoints, are approaches that reinforce the interest of locoregional treatment by radiotherapy [98].

## 4. Conclusions

As our knowledge increases in understanding the biological mechanisms of NSCLC resistance to chemoradiotherapy, we can assess some strategies to improve the survival of patients. New radiosensitizing treatments have emerged recently such as PARPi that might improve the response to chemoradiotherapy in NSCLCs [68], but efficacity seems limited. New TKI targeting specific mutated NSCLCs could play an important role in association with chemoradiotherapy [8,9,10,41,49], but we must be careful due to the potential increase in lung toxicities [99], particularly with use of adjuvant or concomitant anti PD-1 immunotherapies [100]. Furthermore, TKI used as treatments for NSCLC currently target only driver oncogenes in a limited number of patients (less than 50%) [101]. The antitumor immune system is essential to control cancer [102], and immunotherapy is now a standard treatment in NSCLCs [52]. Adjuvant immunotherapy can induce more lung toxicities after chemoradiotherapy [3], questioning the possibility of its concomitant association. New modalities of radiotherapy such as proton irradiation sparing more healthy lung or heart tissues could be promising to combine radiotherapy and immunotherapy [92,103,104]. Escalating the photon or ion irradiation dose in a limited hypoxic area guided by PET imaging might also represent an interesting strategy to increase efficacity on the CSCs and improve tumor control [60,94,105]. Taken together, all these strategies give hope to improve survival for unresectable stage III NSCLC patients in the near future.

## Figures and Tables

**Table 1 cancers-14-02829-t001:** Clinical trials in unresectable stage III NSCLCs treated by radiotherapy with concomitant or induction checkpoint inhibitor immunotherapy.

NCT Number	Acronym	Status	Induction Drug	Concomitant Drug with Radiotherapy	Consolidation Drug	Phases	Enrollment
NCT04765709	BRIDGE	Not yet recruiting	Durvalumab ^1^ plus platinum-based chemotherapy	Durvalumab	Durvalumab	Phase 2	65
NCT02434081	NICOLAS	Completed	None	Nivolumab ^2^ plus platinum-based chemotherapy	Nivolumab	Phase 2	94
NCT04577638	AIRING	Recruiting	None	Nivolumab *	Nivolumab	Phase 2	60
NCT04003246		Active, not recruiting	None	Durvalumab	Durvalumab	Phase 2	50
NCT04230408	PACIFIC BRAZIL	Recruiting	Durvalumab plus platinum-based chemotherapy	Durvalumab plus platinum-based chemotherapy	Durvalumab	Phase 2	48
NCT03631784	KEYNOTE-799	Active, not recruiting	Pembrolizumab ^2^ plus platinum-based chemotherapy	Pembrolizumab plus platinum-based chemotherapy	Pembrolizumab	Phase 2	217
NCT04085250		Recruiting	Nivolumab plus platinum-based chemotherapy	Nivolumab plus platinum-based chemotherapy	Nivolumab or observation	Phase 2	264
NCT04982549		Recruiting	None	Durvalumab plus platinum-based chemotherapy	Durvalumab	Phase 2	35
NCT03840902		Active, not recruiting		Bintrafusp alfa M7824 ^1^ (bifunctional fusion protein composed of a mAb against PD-L1) plus platinum-based chemotherapy	Bintrafusp alfa M7824	Phase 2	168
NCT04364048		Recruiting	Durvalumab	platinum-based chemotherapy	Durvalumab	Phase 2	54
NCT04092283		Recruiting	None	Durvalumab plus platinum-based chemotherapy	Durvalumab	Phase 3	660
NCT05128630	DEDALUS	Recruiting	Durvalumab plus platinum-based chemotherapy	Durvalumab (with hypofractionated radiation therapy)	Durvalumab	Phase 2	45
NCT02621398		Active, not recruiting	None	Pembrolizumab plus platinum-based chemotherapy	Pembrolizumab	Phase 1	30
NCT04202809	ESPADURVA	Recruiting	None	Durvalumab plus platinum-based chemotherapy	Durvalumab	Phase 2	90
NCT03801902		Active, not recruiting	Durvalumab (2 weeks before RT)	Durvalumab	Durvalumab	Phase 1	22
NCT04372927	ADMIRAL	Recruiting	Durvalumab plus platinum-based chemotherapy	Durvalumab plus platinum-based chemotherapy (with Hypofractionated Radiation Therapy)	Durvalumab	Phase 2	40
NCT03999710		Recruiting	Sequential chemotherapy	Durvalumab	Durvalumab	Phase 2	53
NCT04776447	APOLO	Recruiting	Atezolizumab ^1^	Platinum-based chemotherapy	Atezolizumab	Phase 2	51
NCT04013542		Recruiting	None	Ipilimumab ^3^ and Nivolumab	Ipilimumab and Nivolumab	Phase 1	20
NCT04026412	ChekMate 73L	Recruiting	None	Ipilimumab and Nivolumab or Nivolumab alone/plus platinum-based chemotherapy	Nivolumab	Phase 3	888
NCT05298423		Not yet recruiting	None	Pembrolizumab and Vibostolimab ^4^ plus platinum-based chemotherapy	Pembrolizumaband Vibostolimab	Phase 3	784
NCT04380636		Recruiting	None	Pembrolizumab plus platinum-based chemotherapy	Pembrolizumab and Olaparib ^5^ or placebo	Phase 3	870

^1^: Anti PD-L1 immunotherapy (check point inhibitor). ^2^: Anti PD-1 immunotherapy (check point inhibitor). ^3^: Anti cytotoxic T-lymphocyte-associated protein 4 (CTLA-4) immunotherapy (check point inhibitor). ^4^: Anti–T-cell immunoreceptor with Ig and ITIM domains (TIGIT) antibody (check point inhibitor). ^5^: Olaparib is a Poly(ADP-ribose) polymerase inhibitor (PARPi). PARPi exert an antitumor activity by inhibiting DNA repair pathway. These treatments are used with chemotherapy and are investigated in combination with immunotherapy. * Patients not eligible to concomitant chemotherapy. Data from Clinicaltrials.gov (accessed on 1 June 2022) with key words: Non small cell lung cancer stage 3; Radiotherapy. Only recent studies referring to interventional treatment of checkpoint inhibitor immunotherapies used in induction or concomitant with radiotherapy in unresectable stage III NSCLC were listed.

**Table 2 cancers-14-02829-t002:** Trials in unresectable stage III NSCLCs treated by protontherapy.

NCT Number	Status	Interventions	Phases	Enrollment	Results
NCT00495170	Completed	PT with concomitant chemotherapy (Carboplatin + Paclitaxel)	Phase 2	64	Median OS was 26.5 months. Rates of grade 2 and 3 acute esophagitis were 18 (28%) and 5 (8%), respectively. Acute grade 2 pneumonitis occurred in one (2%) patient.
		74 Gy (RBE) 2 Gy/fraction for 37 fractions			
NCT01993810	Recruiting	Arm 1: RT with concomitant chemotherapy	Phase 3	330	
		Arm 2: PT with concomitant chemotherapy (Carboplatin + Paclitaxel)			
NCT01629498	Recruiting	Arm 1: IMRT with concomitant chemotherapy	Phase 1|Phase 2	100	
		Arm 2: IMPT with concomitant chemotherapy			
NCT01770418	Active, not recruiting	PT with concomitant chemotherapy: Dose Level 1: 60 Gy (RBE) at 2.5 Gy(RBE) per fraction × 24 fractions Dose Level 2: 60 Gy (RBE) at 3 Gy (RBE) per fraction × 20 fractions Dose Level 3: 60.01 Gy (RBE) at 3.53 Gy (RBE) per fraction × 17 fractions Dose Level 4: 60 Gy (RBE) at 4 Gy (RBE) per fraction × 15 fractions	Phase 1|Phase 2	32	
NCT02172846	Completed	Hypofractionated PT (60 Gy in 15 fractions) with concomitant chemotherapy (Carboplatin + Paclitaxel)	Phase 1	23	Acute grade 2 esophagitis in seven patients (35%) and grade 2 pneumonitis in one patient (5%).
NCT04432142	Recruiting	Cohort one: RT with concomitant chemotherapy and Durvalumab in consolidation treatment	Observational: Immune changes induced by PT or RT	80	
		Cohort two: PT with concomitant chemotherapy and Durvalumab in consolidation treatment			

IMPT = Intensity-Modulated Protontherapy; PT = protontherapy; RBE = relative biological effectiveness; RT = radiotherapy with photons X. Data from Clinicaltrials.gov (accessed on 1 June 2022) with key words: Non small cell lung cancer stage 3; Radiotherapy or Protontherapy.

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
