# Peer review of "Radioresistance of Non-Small Cell Lung Cancers and Therapeutic Perspectives"

_cancers, 2022, doi:10.3390/cancers14122829_

Round 1

Reviewer 1 Report

Review of the paper titled: “Radioresistance of Non-Small Cell Lung Cancers and Thera-2 peutic Perspectives “ by Mathieu Césaire and colleagues.

Comments

Thank you for giving me the chance to read and review this paper. In this review the authors focused on biological mechanisms of stage III NSCLC radio-resistance and then explored some strategies to overcome it with new drugs or radiotherapy modalities.

The review is well written and includes the most important available literature so far. To make reproducible the authors methods used in writing the paper, I would suggest adding a methods chapter, mentioning the research method. 

Author Response

In order to write and structure this review article, the authors shared the writing process according to their own expertise, mainly using pubmed website and other bibliographic sources. No systematic method was used with several research engines. A selection of appropriate articles to be cited was done according to the context and to the suitability of the selected publications. The literature in our topic is very large and several filtrations needed to be done in order to extract a pertinent information. Therefore, we consider that a method section is not mandatory for this review article.

Reviewer 2 Report

The manuscript by Cesaire et al entitled " Radioresistance of Non-Small Cell lung Cancers and Therapeutic Perspectives" provide a detailed description of biological mechanisms involved in chemoradioresistance and strategies to overcome this in NSCLC. The review is well-written and includes a number of important and recent clinical trials in the field. 

1) Some typos in the text are found in lines 44, 220, 393, 409.

Author Response

Thank you for the comments, we modified the typos accordingly

Reviewer 3 Report

I had the privilege to read a review the Review of cesaires et al. with "Radioresistance of Non-Small Cell Lung Cancers and Therapeutic Perspectives"

1)no new results have been reported in this paper;
2) this review is, in my opinion, quite long and its length undermines the readability of the whole paper.

Author Response

1) as the submitted manuscript is a review article, we agree that no new results are reported, but we hope that such review of the litherature will be of interest for the community.

2) we tried to be as synthetic as possible, but we agree that the length of this review article could be a bit long for readers. In agreement with the editor, we decided to include a new table with clinical trials in order to illustrate the text and to improve the overall quality of the article.